# Clinician Perceptions of Barriers and Facilitators for Delivering Early Integrated Palliative Care via Telehealth

**DOI:** 10.3390/cancers15225340

**Published:** 2023-11-09

**Authors:** Katrina Grace Sadang, Joely A. Centracchio, Yael Turk, Elyse Park, Josephine L. Feliciano, Isaac S. Chua, Leslie Blackhall, Maria J. Silveira, Stacy M. Fischer, Michael Rabow, Finly Zachariah, Carl Grey, Toby C. Campbell, Jacob Strand, Jennifer S. Temel, Joseph A. Greer

**Affiliations:** 1Harvard T. H. Chan School of Public Health, Boston, MA 02115, USA; kasadang@lifelongmedical.org; 2Lifelong Medical Care Family Medicine Residency, Richmond, CA 94801, USA; 3Massachusetts General Hospital, Boston, MA 02114, USA; jcentracchio@miami.edu (J.A.C.); yaelturk1@gmail.com (Y.T.); epark@mgh.harvard.edu (E.P.); jtemel@mgh.harvard.edu (J.S.T.); 4The Johns Hopkins University School of Medicine, Baltimore, MD 21205, USA; jfelici4@jhmi.edu; 5Brigham and Women’s Hospital & Dana-Farber Cancer Institute, Boston, MA 02215, USA; ichua@bwh.harvard.edu; 6Department of Palliative Care, University of Virginia, Charlottesville, VA 22903, USA; lb9x@uvahealth.org; 7Department of Geriatrics and Palliative Medicine, Ann Arbor Veterans Affairs (VA) Medical Center, University of Michigan, Ann Arbor, MI 48104, USA; mariajs@med.umich.edu; 8University of Colorado Hospital, Aurora, CO 80045, USA; stacy.fischer@cuanschutz.edu; 9University of California San Francisco Medical Center, San Francisco, CA 94143, USA; mike.rabow@ucsf.edu; 10City of Hope, Duarte, CA 91010, USA; fzachariah@coh.org; 11Wake Forest Baptist Health, Winston-Salem, NC 27157, USA; cgrey@wakehealth.edu; 12Department of Hematology/Oncology and Palliative Care, University of Wisconsin School of Medicine and Public Health, Madison, WI 53705, USA; tcc@medicine.wisc.edu; 13Mayo Clinic, Rochester, MN 55905, USA; strand.jacob@mayo.edu

**Keywords:** palliative care, telehealth, implementation, cancer

## Abstract

**Simple Summary:**

For patients with a life-limiting illness like advanced cancer, early engagement with palliative care services in the ambulatory care setting can provide several benefits, including improved symptom management and quality of life. Telehealth may be a viable way to deliver these services, especially for populations with limited access to palliative care in a traditional outpatient clinic. We surveyed clinicians who provide palliative care services to patients with advanced lung cancer regarding the barriers, facilitators, and benefits of using telehealth for delivering early integrated palliative care. Our findings show that policies and interventions targeting patient-, organizational-, and systems-based levels are needed to support the use of telehealth for palliative care.

**Abstract:**

Early integrated palliative care (EIPC) significantly improves clinical outcomes for patients with advanced cancer. Telehealth may be a useful tool to deliver EIPC sustainably and equitably. Palliative care clinicians completed a survey regarding their perceptions of the barriers, facilitators, and benefits of using telehealth video visits for delivering EIPC for patients with advanced lung cancer. Forty-eight clinicians across 22 cancer centers completed the survey between May and July 2022. Most (91.7%) agreed that telehealth increases access to EIPC and simplifies the process for patients to receive EIPC (79.2%). Clinicians noted that the elderly, those in rural areas, and those with less-resourced backgrounds have greater difficulty using telehealth. Perceived barriers were largely patient-based factors, including technological literacy, internet and device availability, and patient preferences. Clinicians agreed that several organizational factors facilitated telehealth EIPC delivery, including technological infrastructure (85.4%), training (83.3%), and support from study coordinators (81.3%). Other barriers included systems-based factors, such as insurance reimbursement and out-of-state coverage restrictions. Patient-, organization-, and systems-based factors are all important to providing and improving access to telehealth EIPC services. Further research is needed to investigate the efficacy of telehealth EIPC and how policies and interventions may improve access to and dissemination of this care modality.

## 1. Introduction

Palliative care significantly improves clinical outcomes, such as quality of life, symptom burden, and mood, for patients with serious illnesses [1,2,3]. Traditionally, palliative care has been utilized predominantly in the inpatient setting and generally late in the disease course [4]. However, over the past 15 years, palliative care delivery models have evolved in the setting of research showing that incorporating longitudinal palliative care early in the disease course and in the outpatient setting, also known as “early integrated palliative care” (EIPC), can further improve clinical outcomes, particularly for patients with advanced cancer. Randomized trials testing the efficacy of EIPC for patients with advanced (i.e., stage IV) or newly diagnosed cancer have demonstrated not only improvements in patient-reported quality of life and mood, but also earlier hospice enrollment and increased discussion of end-of-life wishes between patients and their oncologists [5,6]. Furthermore, Hui and colleagues found that patients with cancer who were referred to outpatient palliative care had fewer hospitalizations and less aggressive end-of-life care compared to those who only received inpatient palliative care [7]. Based on these data and supporting literature, the American Society of Clinical Oncology, Institute of Medicine, and National Comprehensive Cancer Network have recommended EIPC for any patient with advanced cancer [8,9,10].

Patients with advanced cancer have frequent appointments for oncology care, including clinic visits to consult with the oncologist, receive systemic therapy, and undergo laboratory testing and radiographic imaging, which can be burdensome. To minimize patient burden, patients receiving EIPC often meet with both their oncologist and palliative care clinicians on the same day. However, palliative care programs at cancer centers vary widely in terms of their resources [11], and most face shortages in palliative care clinicians and support staff [12], limiting the delivery of EIPC in line with national guidelines. Additionally, some palliative outpatient clinics are not embedded at their institution’s cancer center, making it even more burdensome for patients with cancer. A sustainable, patient-centered EIPC delivery system must strive to minimize patient trips and consolidate appointments, including conducting joint palliative–oncology visits when appropriate and feasible. Home-based palliative care has been developed as an alternative to services at healthcare facilities, but this model also requires a large, if not larger, number of clinicians to provide care and is not feasible for remote areas [13]. Investigating alternative modalities for EIPC delivery is essential to increasing access to and utilization of limited palliative care resources in a patient-centered model. Addressing this need is especially imperative in the context of palliative care disparities that disproportionately affect particular communities such as racial and ethnic minorities [14], immigrants [15], and patients living in remote, resource-poor, and/or rural areas [16].

More recently, investigators have begun to examine telehealth as a mechanism for EIPC delivery. Over the past few decades, telehealth video services have expanded to care for patients with various chronic health conditions due to growing interest among stakeholders, including physician organizations, healthcare systems, and insurance providers [17,18]. Interest in and use of telehealth grew tremendously in the setting of the COVID-19 pandemic [19]. Studies of the use of telehealth video house calls for managing health conditions such as Parkinson’s disease and diabetes have demonstrated the feasibility, acceptability, and preliminary efficacy of this delivery model [20,21]. Although limited, studies of telehealth video services for palliative care show similar promise [22,23]. Perri and colleagues found that, for residents in long-term care facilities, telehealth video conferencing was a feasible means of providing palliative care and most clinical staff and families were satisfied with the service [24]. These findings suggest that telehealth video services can be a viable delivery model for EIPC.

In two systematic reviews, researchers identified multiple barriers to adopting telehealth services across organizational, provider, and patient levels [25,26]. Such challenges likely also apply to the delivery of EIPC using telehealth video services. For the current study, we aimed to investigate barriers and facilitators to implementing EIPC via telehealth by surveying palliative care clinicians participating in a large-scale multisite palliative care trial comparing in-person and video visit-based services. We also sought to explore clinician perceptions of whether these barriers differ for patient subpopulations as well as identify any interventions that clinicians and support staff have used to overcome these barriers.

## 2. Materials and Methods

### 2.1. Sample

The sample for this survey consisted of palliative care and oncology clinicians who served as study interventionists for a multisite randomized trial comparing the effectiveness of delivering EIPC via video visits versus in-person in clinic for patients with advanced lung cancer. The protocol procedures for this trial have been published previously [27]. Participating sites included 22 academic cancer centers throughout the U.S. All study clinicians were advanced practice providers or board-certified physicians specializing in palliative care and/or oncology who delivered at least monthly EIPC intervention visits via secure video and/or in person to enrolled patients, beginning within 12 weeks of diagnosis of advanced lung cancer.

### 2.2. Survey Development and Analyses

A multidisciplinary team of clinical investigators specializing in palliative care, oncology, and clinical psychology developed a 36-item survey to evaluate the implementation of delivering EIPC via telehealth using secure video technology. The survey was designed based on the Consolidated Framework for Implementation Research (CFIR), assessing salient constructs related to intervention characteristics, outer setting, inner setting, characteristics of individuals, and process. CFIR is a well-established framework for evaluating and enhancing real-world implementation of healthcare delivery interventions [28,29]. This one-time survey focused on clinician perceptions regarding the benefits, barriers, and facilitators in using telehealth video visits for delivering EIPC. Several items also specifically addressed challenges in implementing equitable telehealth palliative care for different subpopulations and any strategies that the study sites used to address those challenges. The survey included 31 quantitative items scored on a five-point rating scale ranging from “disagree” to “agree” as well as five free text qualitative questions in which respondents wrote in their perceptions regarding the barriers, facilitators, and beneficial impact of delivering EIPC via telehealth (Appendix A Table A1). Participants also completed a brief sociodemographic questionnaire that asked about their age, gender, race, ethnicity, medical specialty, and years of practice. The Dana-Farber/Harvard Cancer Center Institutional Review Board deemed the study protocol and survey exempt.

Study staff sent the electronic survey via Research Electronic Data Capture (REDCap) REDCap Consortium, Nashville, TN, USA to all the palliative care clinicians serving as study interventionists across 22 cancer centers participating in the large-scale randomized trial of EIPC delivered via telehealth versus in-person in clinic. The survey was originally sent on 5/23/22 with subsequent reminders on 6/6/22, 6/20/22, and 7/13/22. See Table A1 in the appendix for the full survey.

### 2.3. Analyses

For the quantitative survey items, we used descriptive statistics to summarize clinician perceptions of the benefits, barriers, and facilitators for delivering EIPC via telehealth. We collapsed the five-point scale into three categories: “Disagree”, “Neither Agree nor Disagree”, and “Agree”. For the free text items, the study team employed rapid qualitative analysis methods to develop an initial coding scheme that aligned with the survey questions regarding barriers and facilitators for delivering EIPC via telehealth video visits [30]. Two independent coders (J.A.C., Y.T.) then reviewed all participant comments to identify key themes along with illustrative quotes. The study team reviewed any free text responses in which the coders disagreed in order to come to a consensus.

## 3. Results

### 3.1. Participant Characteristics

Ninety-four clinicians received the survey, and forty-eight (51.1%) responded. Table 1 describes the sociodemographic characteristics of the respondents. Most of the respondents were female (64.6%), white (85.4%), and not Hispanic or Latino/a/x (93.7%). A total of 75% of respondents were medical doctors and the remaining 25.0% were advanced practice providers (i.e., nurse practitioners or physician assistants), with 62.5% reporting being in practice for more than 10 years.

### 3.2. Clinician Attitudes Regarding Telehealth Palliative Care

Clinician attitudes toward palliative care delivery and telehealth are depicted in Table 2. Most (91.7%) clinicians agreed that telehealth increases access to EIPC for patients with advanced lung cancer and that telehealth simplifies the process for patients to receive EIPC (79.2%). The majority (81.3%) also agreed that telehealth has increased access to EIPC for patients who would not have otherwise have access. However, 66.0% agreed with the survey statement that “most patients with advanced lung cancer will agree to telehealth to receive EIPC”. Most clinicians reported being confident in their ability to use telehealth to deliver EIPC (93.8%).

In the open-ended query asking which particular patients had difficulty using telehealth, respondents cited older patients, individuals who lived in rural areas, and individuals from lower-resourced/socioeconomic backgrounds. One clinician explained, “it’s more likely that an elderly (75+ yo) patient struggles with telehealth”. Respondents described how patients with limited technology or health literacy had difficulties with telehealth, which resulted in the patients’ dislike or lack of use of the patient portal or web-based clinic access. One clinician shared, “Those with poor technology literacy (regardless of age) are often relying on provider initiation of visits via email or text message invitations, rather than using the app- or web-based access that is part of our general outpatient workflow”. Individuals with vision or hearing impairments, limited English proficiency, psychiatric illness, or advanced medical disease also had challenges. Clinicians also noted concerns for individuals with fewer resources, such as personal technological devices or established caregiver support systems.

### 3.3. Clinician Perceptions of Barriers to Telehealth Palliative Care Delivery

Clinician perceptions of barriers to telehealth palliative care delivery are shown in Table 3. Notably, 41.7% agreed that logistical challenges to providing EIPC via telehealth exist. In terms of barriers, many respondents agreed that lack of patient broadband or internet availability (68.8%) and lack of patient access to devices (60.4%) impacted their ability to provide telehealth for palliative care. Other frequently reported barriers included workflow or scheduling issues (35.4%) and staffing issues (39.6%).

Aside from patient characteristics that were associated with difficulties using telehealth, clinicians also reported challenges with patients’ demeanor during telehealth visits. In particular, they shared that patients tended to be less focused and engaged, and, at times, overly casual when they were on telehealth visits. One clinician explained, “[…] Some are distracted and unable to focus sufficiently to have a meaningful encounter”, and that patients behave differently during telehealth encounters as opposed to how they might approach in-person visits. For example, patients and their families could be seen “grocery shopping, at concerts, in bathrooms for scheduled visits—or they might ask providers to callback at more convenient time, etc.”

Clinicians also cited institutional barriers to telehealth visits, including the need for availability of Health Insurance Portability and Accountability Act of 1996 (HIPAA)-compliant telehealth stations, training, and leadership support. Within hospitals, there are an insufficient number of telehealth stations, and clinicians noted needing to acclimate to various telehealth platforms. One clinician explained, “We have implemented multiple telehealth platforms …This resulted in some delays related to the transitions from one platform to the next as patients and staff acclimated to the new technology”. One respondent reflected on “the lack of leadership at the highest levels of the cancer center to support the delivery of telehealth” in which the clinician called for a “[…] greater alignment between cancer center leadership, informatics department, administrative support, and clinic leadership before telehealth palliative care is implemented smoothly”. External issues included insurance reimbursement for telehealth, out-of-state coverage restriction, and home internet connectivity in rural areas.

### 3.4. Clinician Perceptions of Facilitators to Telehealth Palliative Care Delivery

Facilitators of telehealth palliative care delivery are shown in Table 4. Most clinicians agreed that they had sufficient telehealth training (83.3%), necessary technological infrastructure (85.4%), and ways to share tips on using telehealth amongst each other (83.3%). Clinic planning for telehealth (85.4%) and use of study coordinators (81.3%) were also highly agreed upon facilitators. Fewer clinicians agreed that patients had the necessary technology for telehealth EIPC visits (56.3%).

Clinicians detailed how their healthcare institutions have taken action to facilitate access to telehealth EIPC delivery, largely on an organizational level. Most sites offered support and telehealth education prior to and during telehealth visits. Responses highlighted sites’ efforts to address technological difficulties, such as reverting to phone calls or in-person visits and connecting to alternative HIPAA-compliant platforms. Sites have taken a proactive approach by “[attempting] to identify patients at risk of not joining telehealth” and intervening appropriately. For example, to increase accessibility to telehealth, all sites provided iPads to patients and confirmed internet availability in their respective area. One clinician explained, “We are planning to offer loaner devices for connecting to telehealth appointments as part of the resources provided to study patients”. Palliative care clinicians identified strategies to reduce inequities in access to care such as establishing telehealth at regional clinics to assist patients in connecting with their physicians. Clinician flexibility and patient familial support were also identified as facilitators of equitable palliative care delivery. One clinician explained, “We have coordinated with loved ones/supports when scheduling telemedicine visits”.

### 3.5. Clinician Perceptions of Impacts of Telehealth Palliative Care Delivery

Thematic categories of clinician-described impacts of telehealth palliative care delivery and populations who have benefitted from increased access to EIPC are shown in Table 5. When asked how telehealth has helped to increase access to EIPC at sites, most clinicians reported positive logistical and patient-related impacts. One respondent reflected that “using telehealth to ‘come to [patients]’ adds a layer of convenience that makes it easier for them to get on board with visits”. Respondents felt that both clinicians and patients benefit from telehealth delivery of EIPC. Benefits included the possibility for joint visits with other specialists and family members, increased patient engagement in palliative care, and improved continuity and longitudinal care. Clinicians reported that patients with travel considerations, such as longer commute times and/or limited transportation options, benefitted the most from using telehealth to increase access to EIPC. As one clinician shared, patients who travel extensive distances for their oncology care are both “willing and happy to see [PC clinicians] via telemedicine, and those visits are fruitful to see their home life, activities and social support so early in their disease”. Clinicians also noted that telehealth delivery has increased access to EIPC for patients who refuse to come to clinic for additional appointments or who may be hesitant about meeting with palliative care. Patients with health-related considerations (e.g., difficulty with mobility or frailty), limited support, and comfort with technology were also felt to have improved access to EIPC due to telehealth.

## 4. Discussion

This study demonstrates that, in general, clinicians have favorable attitudes towards telehealth EIPC delivery and note that certain populations struggle with telehealth more than others. Reported barriers to telehealth EIPC delivery were largely patient-related factors and, to a lesser extent, organizational and systems-based factors. Overall, clinicians reported having access to several organizational resources that serve as facilitators of telehealth EIPC delivery. Clinicians also endorsed positive impacts of telehealth EIPC delivery for both patients and institutions as well as identified interventions used to improve access to telehealth EIPC.

Clinicians importantly noted several different populations who have increased difficulty in using telehealth, such as those who are older, experience technological challenges, or have certain sociodemographic characteristics such as low socioeconomic status or being non-English speaking. Patient-based factors were more frequently reported as barriers to telehealth EIPC compared to organizational factors. This finding is consistent with what has been shown in previous literature [25,26,31], especially studies highlighting the importance of patient digital literacy for effective telemedicine care [32]. To our knowledge, no prior research has addressed interventions used to reduce patient barriers and disparities in telehealth EIPC. As part of this study, clinicians reported various patient-directed efforts to improve access to telehealth EIPC. Such strategies included enhancing technological support and workflows, providing patient devices for telehealth, having workflows for using an interpreter, and advocating for policy changes and patient resources such as sufficient internet bandwidth in rural communities. Future research is needed to further characterize which patients would most benefit from telehealth EIPC versus in-person EIPC or in-person management by primary care teams. Additionally, investigators should examine how interventions to address patient barriers can affect access to and efficacy of telehealth EIPC.

Additionally, clinicians identified key organizational factors as facilitators of telehealth EIPC delivery, such as telehealth training, infrastructure, planning, and support staff. These findings are likely due to the rapid expansion of telehealth services nationally and internationally during the COVID-19 pandemic. Policies that have facilitated this expansion include the Centers for Medicare and Medicaid Services’ relaxation of regulations on reimbursement and interstate practice to improve access to telehealth services [33]. State and private insurance providers also made similar changes to improve access [34]. These policy shifts are particularly important because patients with cancer have very high rates of interstate telemedicine cancer care [35]. Because of this expansion on a systems-based level, institutions have likely invested in facilitators reported in this study such as telehealth training, infrastructure, and the development of organizational workflows and policies. However, this landscape will likely change with the end of the COVID-19 pandemic public health emergency, as the U.S. Department of Health and Human Services will thereafter no longer consider telehealth to be an expected benefit [36]. To continue to provide accessible telehealth EIPC services, systems-based changes, such as permanent insurance policy coverage, are needed. Additional studies demonstrating the efficacy of telehealth EIPC services are also needed to support the widespread implementation of such changes.

Clinicians also identified a diverse subset of patients who have benefitted from increased access to EIPC via telehealth services. Some benefits include the ability to maintain continuity of care, conveniences for both patients and providers, and the opportunity for enhanced rapport early in the disease course and over time. These benefits are particularly notable given the existing literature examining rapport building in telehealth services for other medical conditions. For example, in one study of veterans in telehealth pulmonary medicine consultations and another of family therapists conducting telehealth therapy, the authors expressed concerns of how telehealth may impact building rapport with patients and/or caregivers and thus impact outcomes in telehealth [37,38]. Our study adds to the literature that from a provider perspective, telehealth EIPC may have benefits such as equal or enhanced rapport building compared to in-person services. However, further studies are needed to elucidate the patient perspective on receiving EIPC via telehealth.

Our study has several limitations. First, the clinicians who completed the survey are practicing in academic institutions across the nation. Thus, this study may have selection bias given the clinicians had to be part of an existing multisite trial at mostly large academic cancer centers. Perspectives from smaller or non-academic institutions are not captured in this study. Clinician experiences were also limited to the provision of telehealth EIPC for patients with advanced lung cancer, which may impact generalizability to other patient populations who may utilize palliative care. Additionally, since clinicians in this survey were part of a larger research trial, they had additional support with study devices (i.e., iPads) and clinical research coordinators, which is not available in the usual care model. It is unknown if clinic support staff in the real-world setting would have the same capacity or effectiveness as dedicated study personnel. Furthermore, most respondents were white and non-Hispanic or Latino/a/x, which may narrow the representation of clinician experiences. Lastly, as we only collected self-reported survey data, recall bias could have impacted responses.

## 5. Conclusions

Despite the study limitations, the results provide insight into the current state of telehealth EIPC delivery at numerous academic cancer centers across the nation, especially during the rapid expansion of such services due to the COVID-19 pandemic. Clinician responses underscore that patient-, organization-, and system-based factors are all critical to providing and improving access to telehealth EIPC services. We hope this study will encourage further research on the efficacy of telehealth versus in-person EIPC, populations that would most benefit from telehealth EIPC, patient perspectives on telehealth EIPC, and interventions used to improve access to and dissemination of this care model. Given the impending changes for the provision of telehealth services, clinicians and organizations must advocate for healthcare policies that allow for continued insurance coverage of telehealth EIPC services and enhance patient access in numerous ways, such as sufficient internet bandwidth and interstate medical care coverage. Clinicians must also be supported to provide telehealth services with both staff and institutional support and interventions.

## Figures and Tables

**Table 1 cancers-15-05340-t001:** Demographic characteristics of clinician participants (n = 48).

Demographic Variables	n (%)
Age	
18–34	1 (2.1)
35–49	33 (68.7)
50–64	11 (22.9)
Gender	
Female	31 (64.6)
Male	16 (33.3)
Ethnicity	
Not Hispanic or Latino/a/x	45 (93.7)
Hispanic or Latino/a/x	3 (6.3)
Race	
White	41 (85.4)
Asian	5 (10.4)
Black	1 (2.1)
Middle Eastern	1 (2.1)
Current Role	
Palliative Care Clinician	43 (89.6)
Oncologist	2 (4.2)
Oncologist + Palliative Care Clinician	3 (6.3)
Study Role	
Doctor (MD or DO)	36 (75.0)
Advanced Practice Provider	12 (25.0)
Years in practice	
1–10 years	14 (29.2)
11–20 years	20 (41.7)
21–30 years	10 (20.8)
31–40 years	4 (8.3)

**Table 2 cancers-15-05340-t002:** Clinician attitudes regarding EIPC delivery and telehealth (n = 48).

Survey Item	Disagree, n (%)	Neither Agree nor Disagree, n (%)	Agree, n (%)
Telehealth increases access to early integrated palliative care for patients with advanced lung cancer.	0 (0)	4 (8.3)	44 (91.7)
Telehealth simplifies the process for patients to receive early integrated palliative care.	2 (4.2)	8 (16.7)	38 (79.2)
Using telehealth for delivery of early integrated palliative care is cost-saving for: The patient.	7 (14.6)	8 (16.7)	33 (68.8)
Using telehealth for delivery of early integrated palliative care is cost-saving for: The healthcare system.	7 (14.6)	17 (35.4)	24 (50.0)
Telehealth has increased access to early integrated palliative care for patients who would not otherwise have access.	1 (2.1)	8 (16.7)	39 (81.3)
Our clinicians believe that utilizing telehealth to deliver early integrated palliative care is an important part of their job.	2 (4.2)	1 (2.1)	45 (85.4)
Our clinic leadership is committed to utilizing telehealth for delivery of early integrated palliative care.	2 (4.2)	5 (10.4)	41 (85.4)
Our clinicians believe that utilizing telehealth to deliver early integrated palliative care fits well in their current workflow.	7 (14.6)	5 (10.4)	36 (75.0)
Most patients with advanced lung cancer will agree to telehealth to receive early integrated palliative care. *	5 (10.6)	11 (23.4)	31 (66)
Our clinicians are confident in their ability to utilize telehealth to deliver early integrated palliative care.	1 (2.1)	2 (4.2)	45 (93.8)

* Missing n = 1.

**Table 3 cancers-15-05340-t003:** Clinician perceptions of barriers to telehealth EIPC delivery (n = 48).

Survey Item	Disagree, n (%)	Neither Agree nor Disagree, n (%)	Agree, n (%)
Telehealth creates logistical challenges for our clinic to deliver early integrated palliative care.	18 (37.5)	10 (20.8)	20 (41.7)
The multiple steps required for implementation of telehealth has been a challenge for our clinic.	31 (64.6)	5 (10.4)	12 (25)
At my site, patient broadband or internet availability (e.g., poor internet services in rural areas) has impacted our ability to implement telehealth for palliative care.	11 (23.0)	4 (8.3)	33 (68.8)
At my site, lack of insurance coverage/reimbursement has impacted our ability to implement telehealth for palliative care.	21 (43.8)	14 (29.2)	13 (27.1)
At my site, lack of interpreter access and use has impacted our ability to implement telehealth for palliative care.	32 (66.6)	5 (10.4)	11 (23.0)
At my site, lack of access to HIPAA * compliant platforms like Zoom or Doximity has impacted our ability to implement telehealth for palliative care.	46 (95.8)	2 (4.2)	0 (0)
At my site, workflow or scheduling issues has impacted our ability to implement telehealth for palliative care.	29 (60.4)	2 (4.2)	17 (35.4)
At my site, staffing issues (e.g., having medical assistants or research assistants available to contact patients prior to appointments) has impacted our ability to implement telehealth for palliative care.	26 (54.2)	3 (6.3)	19 (39.6)
At my site, provider technological training/issues has impacted our ability to implement telehealth for palliative care.	40 (83.3)	5 (10.4)	3 (6.3)
At my site, lack of patient access to devices has impacted our ability to implement telehealth for palliative care.	13 (27.1)	6 (12.5)	29 (60.4)
At my site, patient difficulty with visual, hearing, or other impairments has impacted our ability to implement telehealth for palliative care.	21 (43.8)	7 (14.6)	20 (41.7)
At my site, lack of caregiver engagement has impacted our ability to implement telehealth for palliative care.	24 (50.0)	13 (27.1)	11 (23.0)

* HIPAA: Health Insurance Portability and Accountability Act of 1996.

**Table 4 cancers-15-05340-t004:** Clinician perceptions of facilitators of telehealth EIPC delivery (n = 48).

Survey Item	Disagree, n (%)	Neither Agree nor Disagree, n (%)	Agree, n (%)
Our patients have access to the necessary technology for telehealth visits to receive early integrated palliative care.	9 (18.7)	12 (25.0)	27 (56.3)
Our clinicians are able to receive assistance with technological issues when utilizing telehealth to deliver early integrated palliative care.	9 (19.1)	8 (17.0)	30 (63.8)
Our clinicians share tips with each other on how best to utilize telehealth to deliver early integrated palliative care.	5 (10.4)	3 (6.3)	40 (83.3)
Our clinicians have had sufficient telehealth training for delivery of early integrated palliative care.	4 (8.3)	4 (8.3)	40 (83.3)
Our clinic has the necessary technological infrastructure to conduct telehealth visits for delivery of early integrated palliative care.	6 (12.5)	1 (2.1)	41 (85.4)
Our clinic planned for how to utilize telehealth to deliver early integrated palliative care.	4 (8.3)	5 (10.4)	39 (81.3)
There is a formally appointed individual within our institution who is responsible for overseeing the implementation of telehealth for delivery of early integrated palliative care.	22 (45.8)	3 (6.3)	23 (47.9)
Study coordinators facilitate our efforts with telehealth for delivery of early integrated palliative care.	6 (12.5)	3 (6.3)	39 (81.3)
Principal Investigator(s) facilitate our efforts with telehealth for delivery of early integrated palliative care.	2 (4.2)	10 (20.8)	36 (75.0)

**Table 5 cancers-15-05340-t005:** Thematic description of impacts of increased access to EIPC via telehealth and populations who benefit.

Major Themes *	Subthemes	Supporting Quotation(s)
Impacts of Increased Access to EIPC via Telehealth		
Patient-related factors	Increased participation and retention Decreased patient energy burden Decreased financial burden Increased tech/telehealth understanding Improves access to PC appts	“Early integration can be tricky as many patients are not yet symptomatic and may not fully appreciate how helpful palliative care can be. Using telehealth to ‘come to them’ adds a layer of convenience that makes it easier for them to get on board with visits”. “Our site has an extremely large urban and suburban catchment area, with many patients having >2-h commute times to campus (one way) for in-person appointments. We cannot always accommodate patients with oncology/infusion and palliative care appointments on the same day. Telehealth allows for better access with less fatigue related to “all-day” time at our institution”.
Logistical factors	Fewer issues with lack of clinic space Scheduling convenience	“[Two] big factors: space (lack of clinic space), [and] energy required to make appointments for patients with serious illness […] all make telehealth palliative care more accessible. The […] space factor [is] more ‘system-related’; the energy expenditure is more related to patient or illness factors”.
Mutual clinician and patient benefit	Joint visits with other specialists and family Increases patient engagement on PC during telehealth appointments Clinicians can follow patients across disease trajectory Increased continuity	“Hard to link with days patients were seeing oncology and provide continuity and talk about sensitive issues. Now we can assure continuity and have control over timing of talking about sensitive issues”. “I’m more productive because I can schedule outside of my regular clinic hours. It is convenient for the patient and the provider alike. I can keep closer tabs on patients that live at a great distance”.
Clinician-related factors	Control over timing to talk about sensitive topics Referrals more convenient for patient care delivery	“Before telehealth, very few of our patients were willing to come to Palliative Care unless the visit was linked with oncology visits, which made scheduling our visit a complex game of Tetris and interfered with clinician continuity; also, due to space limitations, we often had to do visits in non-private infusion settings, which interfered with providing full spectrum pall care. With telehealth, access has improved, continuity has improved, and our schedulers spend far less time re-arranging appointments. As a whole, our clinic is far more efficient and effective with telehealth”.
Populations Who Have Benefitted from Increased Access to EIPC		
Patients with geographic/travel considerations	Patients who live far away/travel far to get to clinic Patients with limited transportation	“[W]e have a large volume of patients who travel long distances for their oncology care. [O]ur limited number of palliative care providers makes it impossible to schedule/coordinate same day appointments with patients’ oncologists. [M]any patients feel palliative care “isn’t necessary” at the early stages of diagnosis, and particularly do not see the benefit of driving 3–4 h just to see us (and I would agree). [T]hey are often willing and happy to see us via telemedicine, and those visits are VERY fruitful to see their home life, activities and social support so early in their disease”.
Patients with scheduling preferences	Patients who will not come into clinic for extra appts Patients hesitant about PC (asymptomatic, never heard of PC) appreciate the convenience Patients with infusion appts that do not coincide with outpatient schedule Patients with fewer infusion appointments	“Some people are more comfortable participating from the comfort of their own home. Enabling them to do so without mandating an in-person visit has increased participation and retention in the study, in my experience”.
Patients with health-related considerations	Patients with physical symptoms	“[PC Telehealth] [a]llows us to treat patients who are frailer and sicker and can’t come to in person visits”.
Patients comfortable with technology	Patients with access to device who are taught platform	“For those who are given access to device and are taught to use the platform”.
Caregiver-related	Patients with limited support	“Very helpful for patients with limited transportation or support”.

* The subtotal for each major theme may contain duplicate free text comments since some comments contained ≥2 codes that could have been classified under separate subthemes.

## Data Availability

The data presented in this study are available upon request from the corresponding author.

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
