# Peer review of "Clinician Perceptions of Barriers and Facilitators for Delivering Early Integrated Palliative Care via Telehealth"

_cancers, 2023, doi:10.3390/cancers15225340_

Round 1
Reviewer 1 Report
Comments and Suggestions for Authors
I have carefully read the manuscript "Clinician Perceptions of Barriers and Facilitators for Delivering Early Integrated Palliative Care via Telehealth" submitted to the journal "Cancers".
This is a study based on a survey of expert clinicians in different cancer centres to find out their views on the benefits and barriers of telemedicine visits for patients with advanced lung cancer.
There is no doubt, especially after the COVID 19 pandemic, that telemedicine is a viable option for providing medical consultations and maintaining continuity of care. However, to say that it is a viable option is not the same as saying that it is an optimal or excellent way of caring for patients.
Introduction
The theoretical framework and the introduction are clear and very well written. The aims of the study are described and justified.
Methods
The method is well developed and appropriate for the study objectives.
I have only one question:
Line F 139 says: ".... on a five-point rating scale from "disagree" to "agree". I assume that the scale ranges from 0 'disagree' to 5 'agree'. As explained below, the scores have been collapsed into three categories: a) disagree; b) neither agree nor disagree; c) agree. This is only possible if the rating score is 6 points (multiple of 3). For this reason, it seems better to speak of a six-point scale from 0 to 5, where 0 means disagree and 5 means agree, with 0 being the first point.
Results
The results are clear and well written.
The acronym HIPAA is mentioned in several paragraphs. It is not clear what it means. I assume it refers to a USA regulation. Can you explain this acronym?
Discussion and conclusion
The discussion is right. It highlights the potential benefits and barriers of telemedicine. It points to patient profiles that may or may not benefit from this modality of care. It would be important to highlight these profiles.
Telemedicine is just one form of care. In my opinion, despite the technological excitement, it cannot replace face-to-face visits. The close contact with the patient cannot be lost, nor can the intangible two-way information that face-to-face visits provide. Therefore, telemedicine can be considered viable, but we cannot claim that it is optimal. Again, it is important to explore the patient profile that can benefit (low palliative complexity, English-speaking, knowledge of and access to information technology, residence far from the care centre...). A key point, in my opinion, that has not been mentioned, is that information and communication technology also allows for an effective connection and support with local health teams (Primary Care) that have the necessary skills to deal with palliative situations with low complexity needs.
This study shows only part of the reality, the opinion of health professionals. It is important to carry out studies to find out what patients think.
Reviewer 2 Report
Comments and Suggestions for Authors
I have read the work very carefully and have no particular criticism to offer. The study is well structured, methodologically sound and presented in a clear and straightforward manner. It represents the outcome of long and demanding work on the part of the authors, whom I can only congratulate. The only thing I would suggest is to go into a little more detail (perhaps with a separate paragraph) on the aspect of digital literacy, by referring to some recent reference studies [Reddy, P., Sharma, B., & Chaudhary, K. (2020). Digital literacy: A review of literature. International Journal of Technoethics (IJT), 11(2), 65-94; Campanozzi, L. L., Gibelli, F., Bailo, P., Nittari, G., Sirignano, A., & Ricci, G. (2023). The role of digital literacy in achieving health equity in the third millennium society: A literature review. Frontiers in Public Health, 11, 1109323; Tinmaz, H., Lee, Y. T., Fanea-Ivanovici, M., & Baber, H. (2022). A systematic review on digital literacy. Smart Learning Environments, 9(1), 1-18).
